# Super-Resolution Reconstruction of Speckle Images of Engineered Bamboo Based on an Attention-Dense Residual Network

**DOI:** 10.3390/s22176693

**Published:** 2022-09-04

**Authors:** Wei Yu, Zheng Liu, Zilong Zhuang, Ying Liu, Xu Wang, Yutu Yang, Binli Gou

**Affiliations:** Jiangsu Co-Innovation Center of Efficient Processing and Utilization of Forest Resources, College of Mechanical and Electronic Engineering, Nanjing Forestry University, Nanjing 210037, China

**Keywords:** engineered bamboo, speckle images, super-resolution, attention-dense residual network

## Abstract

With the global population surge, the consumption of nonrenewable resources and pollution emissions have reached an alarming level. Engineered bamboo is widely used in construction, mechanical and electrical product packaging, and other industries. Its main damage is the material fracture caused by the expansion of initial cracks. In order to accurately detect the length of crack propagation, digital image correlation technology can be used for calculation. At present, the traditional interpolation method is still used in the reconstruction of engineered bamboo speckle images for digital correlation technology, and the performance is relatively lagging. Therefore, this paper proposes a super-resolution reconstruction method of engineering-bamboo speckle images based on an attention-dense residual network. In this study, the residual network is improved by removing the BN layer, using the $L_{1}$ loss function, introducing the attention model, and designing an attention-intensive residual block. An image super-resolution model based on the attention-dense residual network is proposed. Finally, the objective evaluation indexes PSNR and SSIM and subjective evaluation index MOS were used to evaluate the performance of the model. The ADRN method was 29.19 dB, 0.938, and 3.19 points in PSNR, SSIM, and MOS values. Compared to the traditional BICUBIC B-spline interpolation method, the speckle images reconstructed by this model increased by 8.55 dB, 0.323, and 1.43 points, respectively. Compared to the SRResNet method, the speckle images reconstructed by this model were increased by 4.53 dB, 0.111, and 0.14 points, respectively. The reconstructed speckle images of engineered bamboo were clearer, and the image features were more obvious, which could better identify the tip crack position of the engineered bamboo. The results show that the super-resolution reconstruction effect of engineered-bamboo speckle images can be effectively improved by adding the attention mechanism to the residual network. This method has great application value.

## 1. Introduction

Engineered bamboo is a bamboo-based composite material made by the industrialization of raw bamboo, with the advantages of renewable and better mechanical properties [1]. It is an ideal material for construction and electromechanical packaging engineering [2]. However, due to the naturally porous structure of bamboo and the bonding technology of engineered bamboo, crack expansion can lead to damage below the yield limit of the material, thus affecting the load carrying capacity and life of the structure, as well as causing stress softening and stiffness degradation. The tolerance of the material to such crack expansion needs to be assessed by the critical fracture strain energy release rate, and to obtain the critical fracture strain energy release rate accurately, the relationship curve between the fracture strain energy and the crack expansion length must be obtained [3]. The accurate identification of crack tip location and crack extension length has been one of the research difficulties in fracture problems of composite materials. Since the cracks of engineered bamboo are very fine and not easy to identify, the help of a high-performance camera and digital image correlation (DIC) technology can improve the accuracy of identification and provide an effective means to calculate the reliability of the fracture strain energy [4].

The DIC technique is a noncontact measurement method that uses image comparison before and after deformation to detect the deformation and strain distribution on the surface of an object and, thus, measure its mechanical properties. The higher the resolution of an image, the more detail and texture information it contains, and the detail information [5] of an image plays a crucial role in image analysis. In 2017, Pan Bing et al. [6] found that the correlation of the graphs before and after deformation decreased dramatically when the material underwent serious deformation. The problem with using the DIC method is that the pattern of the correlation coefficient distribution graph of the image before deformation is not obvious, which increases the difficulty of the measurement. For this problem, many scholars have proposed inserting intermediate images, a unique tracking strategy combining Fourier–Mellin transform [7] for mutual correlation and reliable guidance, DASIY feature description, and the PatchMatch method [8]; accelerated robust feature algorithms and dependent path passing methods [9]; or a vision-based method for the measurement [10], along with other solutions. However, all suffer from problems such as mismatching or a high computational cost. In practical mechanical measurement experiments, the use of a high-performance ultra-high-speed camera [11] can make the distortion between each frame small enough; however, due to the limitation of camera performance, this problem cannot be solved at the hardware level, and only advanced reconstruction algorithms can be applied to improve the image quality.

Image super-resolution techniques [12] acquire one or more high-resolution images from one or more low-resolution observations. The image quality can be effectively improved by increasing the performance of the image acquisition device; however, there is an upper limit to the number of optical sensors per unit area, so the improvement at the hardware level is limited. In contrast, improving the resolution of the image from the algorithm can better improve the image quality [13]. Traditional algorithms often have problems such as high-frequency information loss and detail blurring during image restoration, and the robustness of the model is poor. After that, people added the idea of deep learning to the super-resolution algorithm and obtained better processing results. The existing super-resolution algorithms based on deep learning include SRCNN, SRResNet, etc. [14]. With the development of hardware devices, the number of layers of neural networks has been deepened, and the problem that the gradient cannot converge has arisen. He Kaiming et al. [15] first introduced a residual structure in ResNet to try to solve the problem of gradient disappearance. Ledig et al. [16] proposed SRResNet, the first application of residual structure to image super-resolution. Zhang Y et al. [17] proposed a residual channel attention network with channel attention mechanism, which is applied to the field of image super-resolution for the first time. Wang Longguang et al. [18] proposed a generic parallax-attention mechanism to capture stereo correspondence, which combines epipolar constraint and an attention mechanism. Saeed Anwar et al. [19] proposed a densely residual Laplacian network, using Laplacian attention to improve the quality of the restored image.

At present, the traditional methods are still used in the reconstruction of engineered-bamboo speckle images for digital correlation technology, and the performance is relatively lagging. The innovations of this study are as follows: (a) the attention module is introduced into the residual network, and an image super-resolution model based on attention-intensive residual network is proposed; (b) the ADRN algorithm is applied to the super-resolution reconstruction of engineered-bamboo speckle images. The contributions of this study are as follows: (a) the super-resolution reconstruction of engineered-bamboo speckle images is realized, and the attention-intensive residual network is applied to the reconstruction of engineering bamboo speckle images for the first time; (b) compared with the traditional method, the quality of the reconstructed image in this study was greatly improved, and the deep information of the speckle images of the engineered bamboo was better excavated.

## 2. Materials and Methods

### 2.1. Imaging

In this experiment, 4-year-old raw bamboo was used as the raw material to prepare engineered-bamboo specimens. The raw material of raw bamboo about 1.7 m above the ground and about 0.3 m in diameter was selected, and the specimens were processed and made by the standard hot-pressing process. The hot-pressing process parameters are shown in Table 1.

The mechanical properties of raw bamboo are not uniform because the fiber content of the green side (outer side) and the matrix content of the yellow side (inner side) are relatively high, so it needs to be soaked and softened and undergo fiber disintegration, drying, sizing, embryo formation, pressurization in all directions, carbonization, cooling and demolding, a surface treatment, pre-cutting, and other processes to form a finished engineered-bamboo specimen with more uniform mechanical properties.

Figure 1 shows the fabricated engineered-bamboo specimens that were used in this study, with a length of 500 mm, a width of 55 mm, a height of 30 mm, and a prefabricated crack length of 160 mm. The number of specimens was 6. Three-point loading experiments were performed on the six specimens in turn, in which the specimens were subjected to shear stresses parallel to the crack surface and perpendicular to the leading edge of the crack to produce a relative slip-open (type II) crack along the direction of the prefabricated crack [20,21].

The experimental equipment included a DDL-100 kN universal testing machine, a 5F08 Thousand Eye Wolf^®^ Revealer high-speed camera, and an image acquisition and parameter control system in Hefei, China. Table 2 shows the types and performance parameters of the experimental equipment.

Using the digital image correlation method to obtain speckle images of engineered bamboo, firstly, we needed to prepare the engineered-bamboo specimens. The specimens were screened and polished, and obvious cracks, bulges, dents, and other defects were removed from the surface. Then, the surfaces of the specimens were polished smooth with sandpaper. Subsequently, the specimens were measured and numbered. A vernier caliper was used to measure the width, thickness, and crack length of the specimens three times, and the average value was taken. A tape ruler was used to measure the length of the specimen three times and the average value was taken. The specimens with width variation exceeding 0.5 mm and thickness variation exceeding 5% of the average value were abandoned. To avoid over-spraying at one time and to ensure that the white matte paint completely covered the base color of the bamboo, the white matte paint was sprayed twice. After the white matte paint dried, black matte paint was sprayed according to the above requirements.

The specimens were then positioned and drawn with three vertical lines perpendicular to the direction of the crack at 25 mm from the left and right ends of the specimen and at the geometric center of the specimen to facilitate specimen installation. Vertical lines were drawn at 310 mm from the beginning of the prefabricated initial crack and horizontal lines were drawn at 5 mm from the upper end of the specimens for subsequent data analysis. Figure 2 shows a schematic diagram of the engineered-bamboo specimens, and Table 3 shows the dimensional parameters of the engineered-bamboo specimens. Finally, the engineered-bamboo speckle specimens were treated to eliminate friction by coating one side of a PVC transparent sheet with olive oil and covering it with another PVC transparent sheet to form a double-layer sheet with oil in the middle. The double-layer sheet was inserted into the original crack with the projected area over the left positioning line to eliminate the influence of contact friction between the upper and lower parts of the initial crack as much as possible during loading in the experiment.

The second step was image acquisition before and after loading. A typical image acquisition system consists of a high-speed camera, a light source, an image acquisition card, and a computer. The specimens were installed so that the centerline of the two supports coincided with the left and right positioning lines, and the loading roller coincided with the center positioning line. The camera was placed directly in front of the specimen, and the camera was placed at a shooting angle perpendicular to the specimen’s speckle surface. The camera’s optical axis was flush with the surface of the specimen, and the lighting source was white light. The relevant parameters in the high-speed camera system control page were set; the sampling frequency of the high-speed camera was set to 20 sheets/s, the loading mode of the universal mechanical testing machine was set to constant loading, and the loading speed was set to 2 mm/min until obvious changes appeared in the test piece cracks or the applied load appeared to drop sharply when the loading was stopped. During the measurement process, speckle patterns on the surface of the tested parts were captured by the camera and stored on a hard disk for the next step of digital image correlation algorithm processing. Figure 3 shows the fracture experimental system of the engineered-bamboo specimens used in this study, Figure 4 shows the experimental process of three-point loading (type II fracture), Figure 5 shows the engineered bamboo speckle specimens, and Figure 6 shows an example of an engineered cracked bamboo speckle image.

### 2.2. Preprocessing Images

The dataset required for the experiment was obtained from the speckle images of engineered-bamboo material taken using the Thousand Eye Wolf high-speed camera of Hefei Fuhuang Junda Hi-Tech Information Technology Co., Ltd. in Hefei, China. There was a total of 1300 images, the length of the images was 4032 pixels, and the width was 1348 pixels. Some pictures from the dataset are shown in Figure 7.

The training set of super-resolution images consisted of corresponding high-resolution and low-resolution image sample pairs. The original high-resolution images were from the original engineered-bamboo speckle image because they have a large unused pure black area, which significantly prolongs the training time. Thus, the original high-resolution speckle image was first preprocessed to remove its black area to obtain the original high-resolution image of the engineered bamboo. Since the resolution of the training-set image was too high, all the images were cropped to obtain 128 × 128 image blocks. The low-resolution images were obtained by the bicubic interpolation down-sampling processing of the high-resolution images, and the proportional coefficient was ×4.

The proportions of the training set, validation set, and test set were at a ratio of 8:1:1, i.e., the number of images in the training set was 1040 and the number of images in both the validation and test sets was 130.

### 2.3. Super-Resolution Reconstruction of Speckle Images of Engineered Bamboo Based on an Attention-Dense Residual Network Model

Today, the interpolation method is almost always used for image reconstruction in engineered-bamboo DIC techniques, and the restored image edge information is not clear enough. To ensure the accuracy of the crack tip detection of engineered bamboo, this paper introduces the ResNet model [22], and it was improved as follows: The batch normalization (BN) layer was removed to improve the training stability of the network and reduce the network-training time. The $L_{1}$ loss function was used to improve the robustness of the network, adding an attention model to improve the network’s attention to image details. Finally, dense connections were used to enable the network to learn more high-frequency features [23].

In this study, a modified attention-dense residual network (ADRN) was applied, and the flow chart of the algorithm is shown in Figure 8, including an input layer, a multilayer attention-dense layer, an up-sampling layer [24], and an output layer.

The input layer consists of a convolutional layer [25] that serves as an input low-resolution image $I^{LR}$, and each image can be represented as a W×H×3 real-valued tensor. The number of output channels of the convolution is C, and the image after the convolution operation can be described as the matrix of W×H×C. The convolution operation maps the information of the three channels of the input image, R, G, and B, to the C convolution kernel components, which deepen the shallow feature information of the input image initially, and then enter the multilayer attention-dense layer afterward.

The multilayer attention-dense layer is divided into a dense residual block and a secondary block, which contains the attention module. This layer can extract deeper feature information from the image and calculate the weight of each channel so that the model can focus more on critical feature information.

The up-sampling layer enlarges the feature map and outputs it to the output layer.

The output layer consists of a convolutional layer that reduces the dimensionality of the feature map to three.

The structure of the image generation algorithm based on the attention residual structure is shown in Figure 9 with three levels. First, the first-level dense residual block consists of n dense residual blocks (n is taken as 16), and each dense residual block contains four cascaded secondary blocks and a convolutional layer. Each secondary block includes a convolutional layer, an activation layer, and an attention module.

#### 2.3.1. Removal of BN Layer

In the super-resolution reconstruction task, the BN layer normalizes the image input to all parts of the network in the network training and suppresses certain colors, textures, and information of the image. However, the super-resolution reconstruction task is very demanding for the image, especially the engineered-bamboo speckle image super-resolution task, which requires increased resolution of the image after reconstruction. At the same time, the color, contrast, brightness, and other detailed information of the original image must not be lost for the tip crack identification work to be carried out smoothly.

In the image super-resolution task, the problems of over-training and over-fitting of the network rarely occur, so the regularization method is also rarely added to the super-resolution reconstruction task, and the regularization role played by the BN layer has a limited effect. At the same time, the BN layer needs to relearn the distribution of the training data in each iteration, which will lead to an increase in the training time cost of the model and generate artifacts, reducing the stability of the network training and the generalization ability of the model.

In order to improve the training speed of the network, reduce the training time, and improve the stability and generalization ability of the network, all BN layers in the network were removed in this study.

#### 2.3.2. Loss Function

In super-resolution tasks, the loss function evaluates the high-resolution image output from the model by $I^{SR}$ relative to the original high-resolution image $I^{HR}$ difference, providing guidance for the model. The most commonly used loss function is the $L_{2}$ loss function, the mean square error (*MSE*), as shown in Equation (1):(1)$$L_{MSE}^{SR}=\frac{1}{r^{2}WH}\overset{rW}{\underset{x=1}{\sum }}\overset{rH}{\underset{y=1}{\sum }}{\left(I_{x,y}^{HR}-I_{x,y}^{SR}\right)}^{2}$$
where $I^{HR}$ and $I^{SR}$ denote the original high-resolution image and the high-resolution image after super-resolution reconstruction, respectively; W and H are the width and height of the image, respectively; and the peak signal-to-noise ratio (PSNR), the image evaluation index, is the ratio of the peak signal intensity to the mean square error, so using the mean square error as the loss function can obtain the maximum PSNR value.

The recently discovered the $L_{1}$ loss function is the mean absolute error (*MAE*), the expression of which is shown in Equation (2):(2)$$L_{MAE}^{SR}=\frac{1}{r^{2}WH}\overset{rW}{\underset{x=1}{\sum }}\overset{rH}{\underset{y=1}{\sum }}\left|I_{x,y}^{HR}-I_{x,y}^{SR}\right|$$

Based on the derivation of its expression, the $L_{2}$ loss function has a stronger penalty for larger errors and a lower penalty for smaller errors. Meanwhile, the $L_{1}$ loss function considers a more uniform error penalty, meaning the $L_{1}$ loss function has better robustness than the $L_{2}$ loss function.

In the specific application of super-resolution reconstruction tasks, the performance of the reconstruction depends more upon the convergence and robustness of the network. The $L_{1}$ loss function and the $L_{2}$ loss function have been compared in the literature [26,27], where different loss functions were chosen in the same network to experimentally test if the $L_{1}$ loss function had stronger convergence than the $L_{2}$. Therefore, the $L_{1}$ loss function was used for training the model in this study.

#### 2.3.3. Choice of Attention Model

The reconstruction of speckle images of engineered bamboo requires high resolution, and crack recognition requires extremely high contrast, texture, and other detailed information. Combining the characteristics of different attention modules in order to enable the model to restore images with not only high resolution but also detailed information, the channel attention structure of the CBAM attention model was added to the network structure after optimization.

#### 2.3.4. Dense Residual Block Design

The current residual blocks used by SRResNet and SRGAN are shown in Figure 10a, the residual block proposed by EDSR is shown in Figure 10b, and the attention-dense residual block designed in this study is shown in Figure 10c. In contrast, the residual block designed in this study removes all BN layers, incorporates the attention module [28], and combines multilevel residual networks and dense connections.

The specific structure of the attention module is shown in Figure 11. After the image passes through the convolution and activation layers, the attention module first performs the average pooling operation and the maximum pooling operation to obtain the feature vectors $F_{avg}^{C}$ and $F_{max}^{C}$, respectively, and then reduces the number of channels by the convolution layer. The convolution kernel size of the convolution layer is X, as shown in Equation (3), at which point, the ratio of the number of channels output to the initial number of channels is M, as shown in Equation (4):(3)$$X=\frac{C}{R}$$
(4)$$M=\frac{1}{R}$$

Among them, *C* is the number of input characteristic channels, and *R* is the typical value. In the reference CBAM structure, the value is 16.

The attention module restores the channels to the initial number of channels through a ReLU activation layer with a convolution kernel size of C to reduce the computational cost of the model. The next step is to calculate the weight size of each channel. At this time, the weight of each channel is more dispersed, and the role of the Sigmoid activation layer is to limit the scale coefficients of each channel between 0 and 1. Finally, the assigned weight coefficients are input to each feature channel to achieve the purpose of making the model pay more attention to the key areas in the image so that the model can reconstruct images with richer details and sharper edges. The calculation process is shown in Equation (5):(5)$$F=\sigma \left(W_{1}\left(ReLU\left(W_{0}\left(F_{avg}^{C}\right)\right)\right)+W_{1}\left(ReLU\left(W_{0}\left(F_{max}^{C}\right)\right)\right)\right)$$
where σ is the Sigmoid activation function, and $F_{avg}^{C}$ and $F_{max}^{C}$ are the feature vectors obtained after the average pooling and maximum pooling operations, $W_{0}\in R^{C/r*C}$ and $W_{1}\in R^{C*C/r}$.

#### 2.3.5. Pseudocode of the Proposed Network

To better explain the procedure of our ADRN, we present the PyTorch-like pseudocode of the ADRN (Algorithm 1). Algorithm 1 specifies the structure of a dense residual block.
**Algorithm 1** The PyTorch-like pseudocode of ADRN.(model): Sequential((0): Conv2d(3, 64, kernel_size=(3, 3), stride=(1, 1), padding=(1, 1))(1): Identity +|Sequential(| (0): ADRN(|  (ADRN1): ResidualDenseBlock_5C(|   (conv1): Sequential(|    (0): Conv2d(64, 32, kernel_size=(3, 3), stride=(1, 1), padding=(1, 1))|    (1): LeakyReLU(negative_slope=0.2, inplace=True)|   )|   (conv2): Sequential(|    (0): Conv2d(96, 32, kernel_size=(3, 3), stride=(1, 1), padding=(1, 1))|    (1): LeakyReLU(negative_slope=0.2, inplace=True)|   )|   (conv3): Sequential(|    (0): Conv2d(128, 32, kernel_size=(3, 3), stride=(1, 1), padding=(1, 1))|    (1): LeakyReLU(negative_slope=0.2, inplace=True)|   )|   (conv4): Sequential(|    (0): Conv2d(160, 32, kernel_size=(3, 3), stride=(1, 1), padding=(1, 1))|    (1): LeakyReLU(negative_slope=0.2, inplace=True)|   )|   (conv5): Sequential(|    (0): Conv2d(192, 64, kernel_size=(3, 3), stride=(1, 1), padding=(1, 1))|   )|  )

## 3. Results

This section may be divided by subheadings. It should provide a concise and precise description of the experimental results, their interpretation, as well as the experimental conclusions that can be drawn.

In order to ensure the fairness and scientific nature of the experiments, the same hardware platform and software environment were used in this study. The hardware platform configuration is shown in Table 4. The experiments were conducted using the Windows 10 64-bit operating system, the programming language was Python 3.7, all deep learning frameworks were built by Pytorch, the IDE was PyCharm Community Edition, and CUDA10.1 and CUDNN7604 were used to accelerate the model training. The parameters of each algorithm network are shown in Table 5.

This study used an initial learning rate of $2\times 10^{-4}$ and the Adam optimizer, with the decay rate parameters set to $\beta_{1}=0.9$ and $\beta_{2}=0.999$. The residual scaling coefficient, calculated before the residuals were added to the main path, was set to 0.2. The ADRN model proposed in this paper must run on GPU to ensure the immediacy of speckle image processing [29].

In order to investigate the impact of the improvement points in this study on network performance, an ablation experiment was used to gradually modify the network model and compare their differences in terms of the objective index PSNR values and the structural similarity (SSIM) values under the condition that the number of images in the test set was 130 and the magnification coefficient was 4. The results are shown in Table 6.

The first step was to replace the $L_{2}$ loss function with the $L_{1}$ loss function. The effect of this improvement is mainly reflected in the improvement in the convergence of network training rather than in the improvement of the value of the evaluation index, which is not very useful for the improvement in network performance.

Secondly, removing all BN layers saved computational resources and memory usage while improving the stability of the network, resulting in small improvements in the PSNR index and SSIM index of 0.33 dB and 0.006, respectively.

Then, the attention mechanism was added to the network structure to enhance the ability of the network to restore detailed critical information. As the attention mechanism added in this study is concerned with channel information, it did not have a positive effect on the PSNR index and even decreased it, but it improved the SSIM index by 0.025, which is a large improvement.

Finally, densely connecting the network allowed the network to learn more high-frequency features with fewer parameters, and it was able to improve the performance of the network while reducing its volume, resulting in a 0.16 dB improvement in the PSNR index and a 0.011 improvement in the SSIM index.

## 4. Discussion

Authors should discuss the results and how they can be interpreted from the perspective of previous studies and of the working hypotheses. The findings and their implications should be discussed in the broadest context possible. Future research directions may also be highlighted.

In order to evaluate the advantages and disadvantages between the improved algorithm and other algorithms in this study, the improved algorithm was compared with other algorithms in terms of the objective index PSNR value and SSIM value and the subjective index mean opinion score (MOS) value under the condition that the number of images in the test set was 130 and the magnification coefficient was 4. Table 7 presents the PSNR and SSIM values and MOS values and the test time of the three algorithms on the dataset of speckle images of engineered bamboo timber.

As can be seen in Table 7, for super-resolution reconstruction of the engineered-bamboo speckle image dataset, the ADRN method was 29.19 dB, 0.938, 3.19 points, and 1.872 s in PSNR, SSIM, MOS, and algorithm time, respectively. The ADRN method improved the PSNR index by 8.55 dB compared to the traditional BICUBIC B-spline interpolation method and improved the SSIM index by 0.323 compared to the traditional method. Moreover, the SSIM value was very close to 1. The effect on this evaluation index was significantly improved, and the improvement in the subjective index MOS value was 1.43 points. This method was significantly better than the traditional interpolation method on the speckle images of engineered bamboo. Its objective evaluation indexes PSNR and SSIM improved by 4.53 dB and 0.111, respectively, relative to SRResNet, which also had a large improvement. There was also partial improvement in the subjective index MOS value, with a value increase of 0.14 points, which confirms the significant role of the attention module and dense residual structure in the image super-resolution reconstruction task. In addition, the test time of the ADRN model is 1.872 s.

The difference between the subjective and objective evaluation indexes was further analyzed. In the objective indexes, both PSNR and SSIM, the ADRN method used in this study was significantly optimized compared to the other two methods, and the reconstructed image quality was greatly improved for the task of super-resolution image reconstruction of engineered bamboo, which better improved the shortcomings of the existing method of restoring images that are not clear enough. In terms of the subjective index MOS, the improvement in ADRN compared to the traditional interpolation method was also significant. However, compared to SRResNet, the improvement was only 0.14 points, which was greatly influenced by human subjective factors in the evaluation. Therefore, the subjective index in the reconstruction quality analysis of engineered-bamboo speckle images is not obvious enough, and further optimization analysis should be carried out.

Figure 12 presents the comparison of the reconstructed images by each algorithm. The low-resolution (LR) image is shown in Figure 12a, the reconstructed image using the BICUBIC B-spline interpolation method is shown in Figure 12b, the reconstructed high-resolution image by the SSResNet method is shown in Figure 12c, and the reconstructed high-resolution image by the ADRN method used in this study is shown in Figure 12d. As can be seen in the figure, the image reconstructed using the BICUBIC B-spline interpolation method has a certain degree of improvement in LR quality compared to the low-resolution image, but the detailed texture information is still very blurred. Meanwhile, the images reconstructed using the SRResNet method and the ADRN method used in this study were significantly improved compared to those reconstructed using the BICUBIC B-spline interpolation method. The edges are significantly sharper, which illustrates the significant effect of deep learning ideas and residual networks on model performance improvement [30]. The method used in this study also shows a small improvement in the effect of the restored images relative to the SRResNet method. The detailed information of the speckle images of the engineered bamboo wood was restored more clearly and the picture features are more obvious, which confirms that the attention mechanism is beneficial for the restoration of detailed features, and, thus, the method proposed in this study has some practicality.

As an innovative aspect of this study, the ADRN algorithm applied the attention module to the dense residual block and constructed the attention-dense residual network as a unit, which can better extract the deep feature information of the speckle images. The network model could focus on the crack location information in the speckle images. This is also the first time that the method was applied to the super-resolution reconstruction of the engineered-bamboo speckle image. The effect of image reconstruction was improved and the practical application value was greater. On the other hand, this study used the common bicubic interpolation down-sampling method to obtain low-resolution images. This degradation process is relatively simple, and the super-resolution network model can only learn the reduction process of the preset degradation process. After that, it can be further studied on paired high-resolution images and low-resolution images that are closer to the real environment.

## 5. Conclusions

In this study, an image super-resolution model based on the attention-dense residual network was proposed to address the current problem that the super-resolution reconstruction of speckle images of engineered bamboo is not clear enough. A large improvement in the quality of the super-resolution reconstruction of speckle images of engineered bamboo was achieved.

In this study, we first produced a speckle image dataset of engineered bamboo and then assigned a training set, validation set, and test set at a ratio of 8:1:1. Then, we designed a multilayer attention-dense module, removed all the BN layers in the network, applied the $L_{1}$ loss function to guide the training of the model, added the attention model to the residual network, combined the residual network step by step to form a multilayer dense structure, and designed and completed the improved attention-dense residual network model. The hardware configuration, software environment, and training parameters of the experiment were given, and the reconstruction results of the engineered bamboo speckle images were finally obtained by the improved model. The performance of each algorithm on the test set was judged in terms of the objective evaluation indexes PSNR and SSIM and the subjective evaluation index MOS, and the results reconstructed from the traditional method and the improved algorithm used in this study on the engineered-bamboo speckle images were compared. The ADRN method was 29.19 dB, 0.938, and 3.19 points in PSNR, SSIM, and MOS values. In the analysis, it was found that the ADRN method proposed in this study improved the objective evaluation indexes PSNR and SSIM and the subjective evaluation index of the image quality by 8.55 dB, 0.323, and 1.43 points, respectively, compared to the traditional BIUCBIU B-spline interpolation method, and by 4.53 dB, 0.111, and 0.14 points, respectively, compared to SRResNet. This method has a very obvious advantage over other methods in the super-resolution reconstruction of speckle images and can restore the image detail information well, amplifying the features of the speckle images of engineered bamboo to better identify the tip crack location. The improved algorithm was proved to be of great value in the task of super-resolution reconstruction of speckle images of engineered bamboo.

In the future, it can be further explored on the acquisition of low-resolution images closer to the real environment. In addition, PSNR, SSIM, and MOS are general evaluation indexes of image, and then other advanced evaluation indexes can be compared and discussed to analyze the quality of super-resolution reconstructed images more comprehensively. Moreover, the method can be applied to speckle image super-resolution reconstruction of other engineered materials, such as wood, to deal with image super-resolution reconstruction tasks quickly and efficiently.

## Figures and Tables

**Figure 1 sensors-22-06693-f001:**
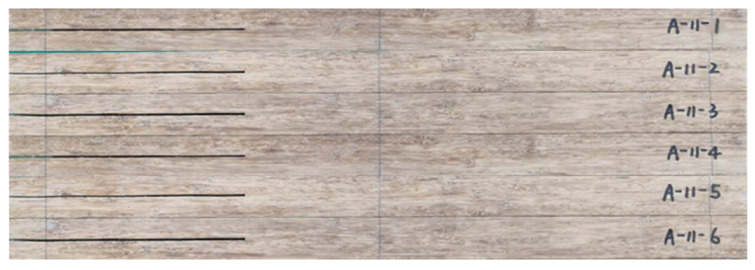
Engineered bamboo specimens.

**Figure 2 sensors-22-06693-f002:**
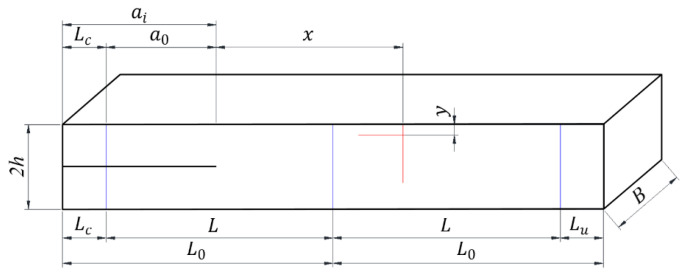
Schematic diagram of engineered-bamboo specimens.

**Figure 3 sensors-22-06693-f003:**
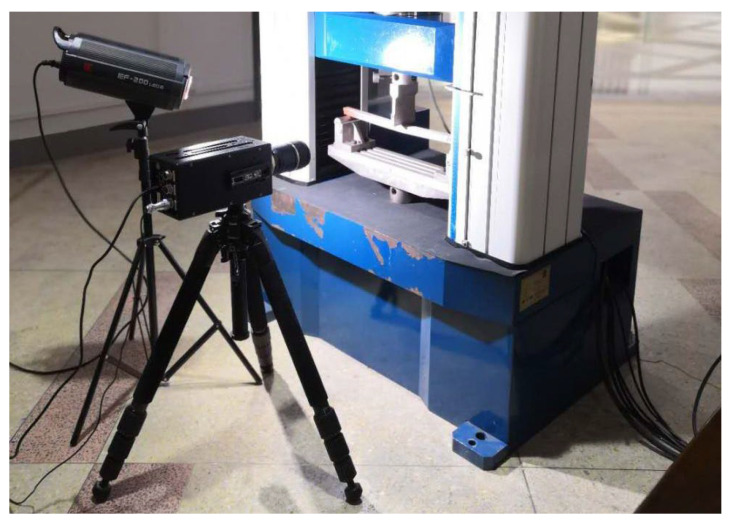
Experimental system for fracture of engineered bamboo using a high-speed camera.

**Figure 4 sensors-22-06693-f004:**
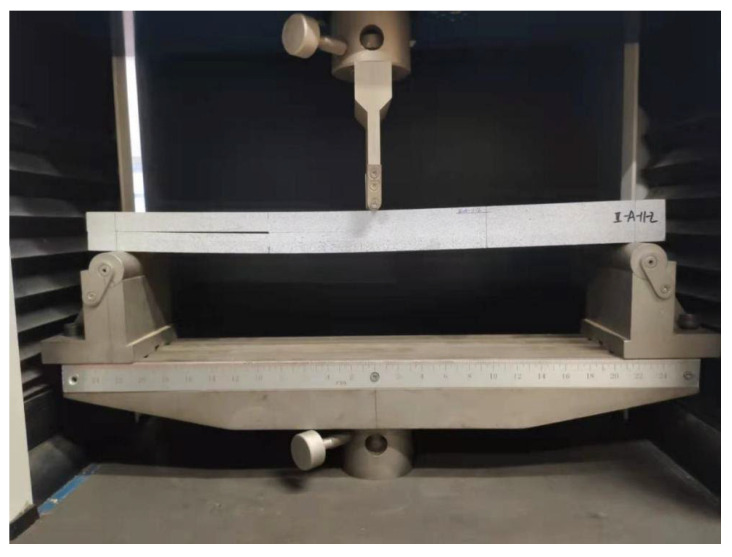
Experimental procedure.

**Figure 5 sensors-22-06693-f005:**
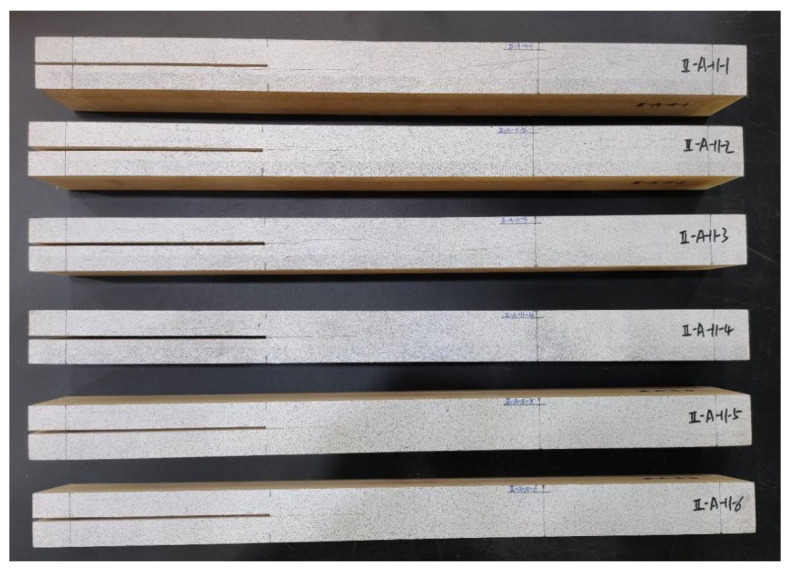
Speckle specimens of engineered bamboo.

**Figure 6 sensors-22-06693-f006:**
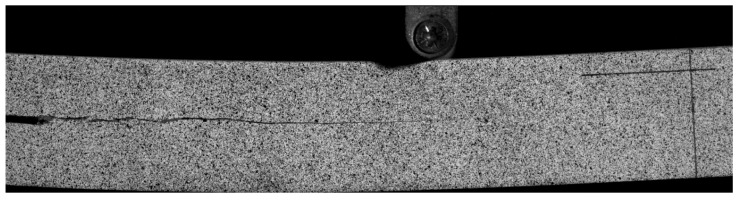
Example of the speckle image of the engineered bamboo crack.

**Figure 7 sensors-22-06693-f007:**
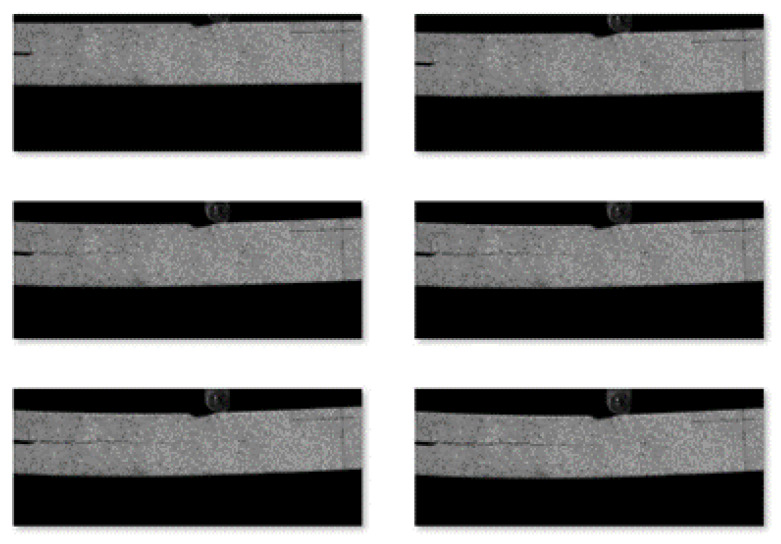
Examples of speckle images of engineered bamboo.

**Figure 8 sensors-22-06693-f008:**
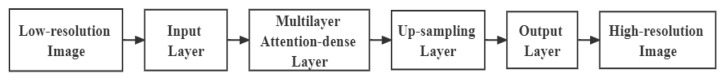
Algorithm flow chart.

**Figure 9 sensors-22-06693-f009:**
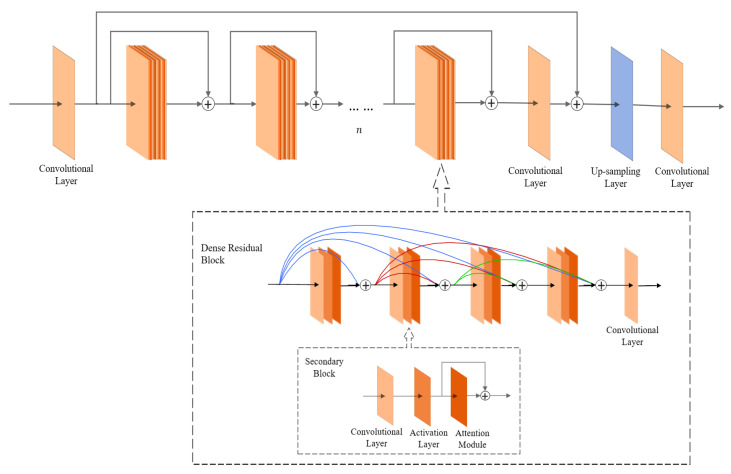
Multilayer attention-dense structure.

**Figure 10 sensors-22-06693-f010:**
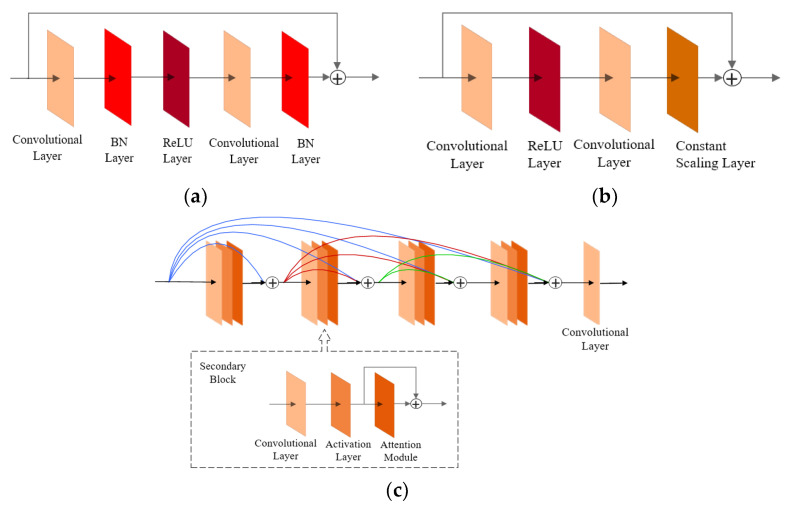
Comparison of residual block structures (**a**) residual blocks used by SRResNet and SRGAN; (**b**) residual blocks proposed by EDSR; (**c**) residual blocks used in this study.

**Figure 11 sensors-22-06693-f011:**
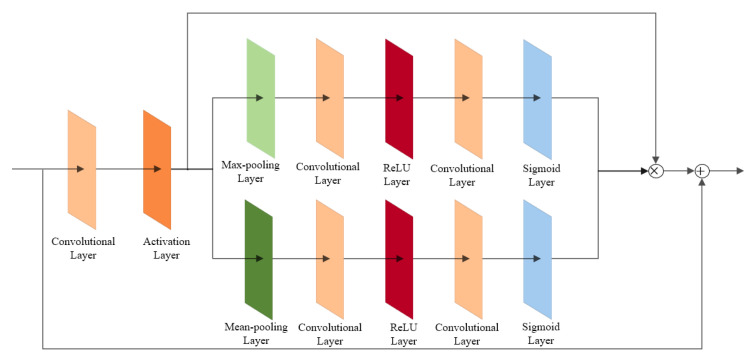
Structure of the attention module.

**Figure 12 sensors-22-06693-f012:**
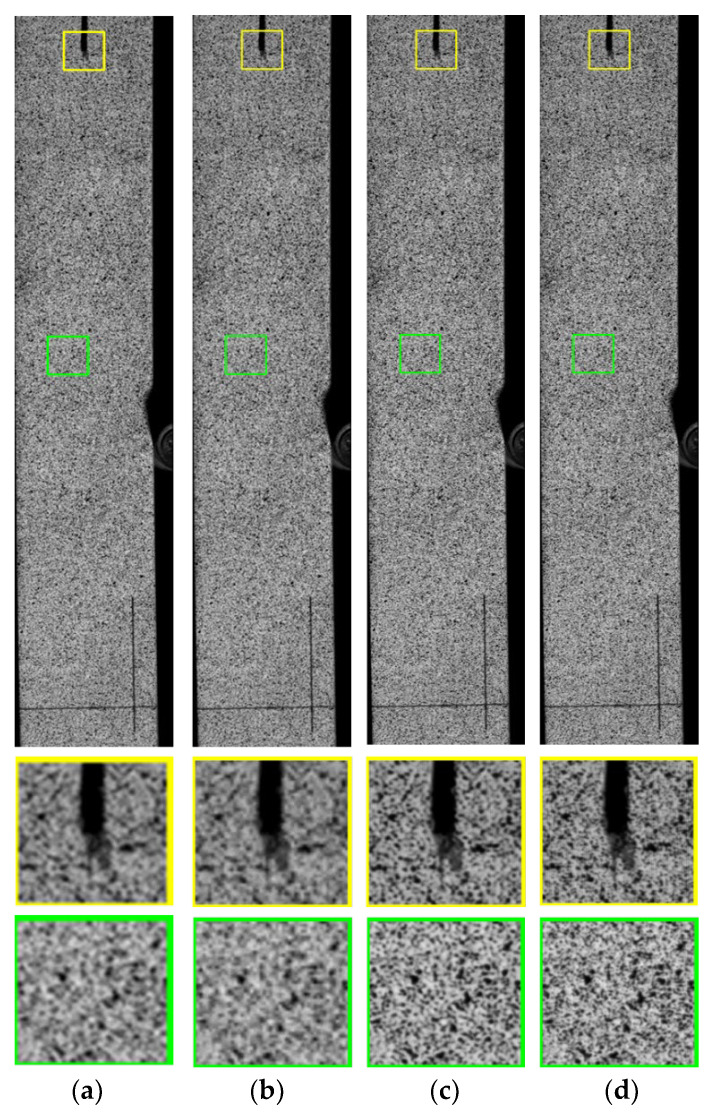
Comparison of reconstructed image effects. (**a**) Low-resolution (LR) image; (**b**) image reconstructed using BICUBIC B-spline interpolation method; (**c**) image reconstructed using SRResNet method; (**d**) image reconstructed using ADRN method.

**Table 1 sensors-22-06693-t001:** Hot-pressing process parameters.

Related Parameters	Glue	Sizing	Press Range	Temperature	Stress	Time
Set value	Phenolic resin	$112\;\mathrm{kg}/m^{3}$	2500 t	135–138 °C	2300 T/SMPa	60 min

**Table 2 sensors-22-06693-t002:** Equipment types and performance parameters.

Equipment Type	Performance Parameters
Universal testing machine	Range: 100 kN Acquisition frequency: 20 Hz
High-speed camera	Maximum resolution: 4000 × 2000 pixels Shooting speed: 4000 × 2000 @ 500 fps Minimum exposure time: 1 µs Image size: 7 µm Sensitivity: 4.64 V/lux.s@525 nm Support trigger method: internal trigger, external trigger
Image acquisition and parameter control system	Acquisition period: 50,000–99,999 µs Magnification: 1× Maximum resolution supported: 4536 × 3024 pixels Supplementary light sources

**Table 3 sensors-22-06693-t003:** Dimensional parameters of engineered-bamboo specimens (unit: mm).

Related Parameters	$a_{0}$	$a_{i}$	L	$L_{0}$	$L_{c}$	$L_{u}$	B	2h	x	y
Numerical value	135	160	225	250	25	25	55	30	150	5

**Table 4 sensors-22-06693-t004:** Hardware platform configuration table.

Hardware Configuration	Name
Processors	Intel Xeon (Xeon) W-2155@3.30 GHz
Motherboard (computer) (lit. lord board)	Dell 0X8DXD Core i7
Video card	Nvidia GeForce GTX 1080 Ti
Video memory	8 G
RAM	Hynix DDR4 2666 MHz 64 G

**Table 5 sensors-22-06693-t005:** Network parameters for each algorithm.

Algorithm	SRResNet	ADRN
Number of residual blocks	32	16
Training image size	128	128
Applicability to pretrained models	deny	deny
Loss function	$L_{2}$	$L_{1}$
Number of feature maps	64	64
Batch size	16	16
Whether or not to add a BN layer	yes	no

**Table 6 sensors-22-06693-t006:** Ablation experiments.

Improvements and Indicators	1st	2nd	3rd	4th	5th
Loss function	$L_{2}$	$L_{1}$	$L_{1}$	$L_{1}$	$L_{1}$
BN layer	√	√	×	×	×
Attention mechanisms	×	×	×	√	√
Dense connection	×	×	×	×	√
PSNR/SSIM	28.92/0.896	28.92/0.896	29.25/0.902	29.03/0.927	29.19/0.938

**Table 7 sensors-22-06693-t007:** Comparison of the mean results of evaluation indexes and test time for the three algorithms on the test set.

Algorithm	PSNR (dB)	SSIM	MOS (Points)	Test Time(s)
BICUBIC B-spline interpolation	20.64	0.615	2.48	1.745 × 10^−4^
SRResNet	24.66	0.827	3.77	1.161
ADRN	29.19	0.938	3.91	1.872

## Data Availability

The data are not publicly available due to the company requirements.

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
