# Peer review of "Super-Resolution Reconstruction of Speckle Images of Engineered Bamboo Based on an Attention-Dense Residual Network"

_sensors, 2022, doi:10.3390/s22176693_

Round 1

Reviewer 1 Report

The paper is well-structured. The novelty should be highlighted and more critical discussion should be provided. Significant improvement is required.

1. More research background of super-resolution reconstruction and imaging processing should be reviewed and discussed e.g. 

  1. Vijay R. Rathod, R. S. Anand & Alaknanda Ashok (2012) Comparative analysis of NDE techniques with image processing, Nondestructive Testing and Evaluation, 27:4, 305-326, DOI: 10.1080/10589759.2011.645820.   2. The novelty and critical discussion should be provided; 3. Challenges and findings of the approach should be highlighted. Further work should be outlined.

Reviewer 2 Report

Super-resolution Reconstruction of Speckle Images of Engineered Bamboo Based on an Attention-dense Residual Network

In this paper, the authors proposed an approach for reconstruction of engineered bamboo speckle images to accurately detect the length of crack propagation. The proposed super-resolution reconstruction method is based on the residual network having an attention-intensive residual block. A comparative analysis of proposed approach and traditional interpolation method is performed by the authors and proposed Attention-dense Residual Network is claimed to be the better. This is an interesting research topic and accurate detection of crack position can be useful for calculating reliability of used material in its commercial applications.

The paper under discussion is well-written and easy to understand. Language of paper is technical and need no further changes. However, following comments need to be addressed:

·      Introductory part is well defined. It covers state of art literature work in a proper way. However, the novelty of the proposed scheme is not discussed at all. Last paragraph of section 1 should discuss the novel contributions.

·      Section 2 discusses the proposed approach, used dataset and experimental dataset in detail. It is necessary to address that why do the authors used Resnet by introducing attention mechanism and why they did not used state-of-the-art vision transformer that already has attention mechanism and outperforming other existing networks.

·      The results section needs to be improved. Instead of just discussion of results, the actual results should also be listed. Like, Performance of proposed approach with or without batch normalization, exact inference time with or without normalizations should be listed in tabular form.

·      Furthermore, instead of comparing proposed approach with traditional method, the proposed algorithm should be compared with state-of-the-art algorithms like Vision Transformer or gMLP.

·      The conclusion section is fine. Please add the future direction part. etc.

Reviewer 3 Report

The abstract should contain information regarding the following points: problem background, materials & methods, attained results, and conclusions.

Numerical results should support conclusions in the abstract.

In the introduction section, the problem statement of the study is not clearly identified.

The literature review is not covering the recent research papers in this field.

Concluding remarks for the reviewed methods are needed at the end of the literature review section.

All the related mathematical equations should be stated, and the associated variables must be identified.

I can see that the proposed method requires numerous inputs. This is undesirable because it will be difficult to implement and tune properly. How do the authors justify that?

It is necessary to add a pseudo-code explaining the proposed algorithm's implementation specifics.

Although the proposed algorithm has advantages, the authors should also state its shortcomings.

The authors should evaluate their results by using some advanced evaluation methods.

The authors should compare their methodology with other contemporary methodologies and highlight why and how their method is better than the existing ones.

The implementation times of the proposed and the comparative algorithms must be given.

The conclusion must reflect the results.

Some linguistic errors are discovered. 

The authors should use more 2021 and 2022 references.

The future works are not presented clearly.

Where can we use this algorithm? please explain its potential real-life applications.

Round 2

Reviewer 1 Report

The improvement is reasonably good. The major contribution and novelty could be further refinement. 

For example, more discussion of research baclground of DIC could be provided e.g. Piotr Kohut, Krzysztof Holak, Adam Martowicz & Tadeusz Uhl (2017) Experimental assessment of rectification algorithm in vision-based deflection measurement system, Nondestructive Testing and Evaluation, 32:2, 200-226, DOI: 10.1080/10589759.2016.1159306.

Author Response

Comment 1.The improvement is reasonably good. The major contribution and novelty could be further refinement. For example, more discussion of research background of DIC could be provided e.g. Piotr Kohut, Krzysztof Holak, Adam Martowicz & Tadeusz Uhl (2017) Experimental assessment of rectification algorithm in vision-based deflection measurement system, Nondestructive Testing and Evaluation, 32:2, 200-226, DOI: 10.1080/10589759.2016.1159306.

Response 1:In order to introduce the DIC background more, this paper added the article ' Experimental assessment of rectification algorithm in vision-based deflection measurement syste ' in Refrence 10, which made the main contribution and innovation of this research in the field of DIC more perfect.

Reviewer 2 Report

Overall, the concerns and recommendations have been addressed. Thanks

Author Response

Thank you very much for your hard review and suggestions, which is very helpful to improve the quality of this paper.

Reviewer 3 Report

I have no further comments

Author Response

Thank you very much for your careful review and suggestions, which is very helpful to improve the quality of this paper.